# Layerwise Bregman Representation Learning of Neural Networks with Applications to Knowledge Distillation

**Ehsan Amid**[1]**, Rohan Anil**[1]**, Christopher Fifty**[2]**, Manfred K. Warmuth**[1]
[1]*Google Research,* [2]*Stanford University*

**Reviewed on OpenReview:** *https://openreview.net/forum?id=6dsvH7pQHH*

## Abstract

We propose a new method for layerwise representation learning of a trained neural network that conforms to the non-linearity of the layer's transfer function. In particular, we form a Bregman divergence based on the convex function induced by the layer's transfer function and construct an extension of the original Bregman PCA formulation by incorporating a mean vector and revising the normalization constraint on the principal directions. These modifications allow exporting the learned representation as a fixed layer with a non-linearity. As an application to knowledge distillation, we cast the learning problem for the student network as predicting the compression coefficients of the teacher's representations, which is then passed as the input to the imported layer. Our empirical findings indicate that our approach is substantially more effective for transferring information between networks than typical teacher-student training that uses the teacher's soft labels.

## 1 Introduction

Representation learning (Bengio et al., 2013) is at the core of Machine Learning (ML) applications. Larger ML models generally extract better explanatory factors from the data, resulting in improved generalization. Such large-scale ML systems train on a vast amount of data. A concrete example is the Ads CTR prediction model (Anil et al., 2022). To effectively learn the intrinsic characteristics of the data (such as user behaviors and seasonal patterns, in our example), such systems rely on training large models with billions of parameters. Yet, such outrageously large models are cumbersome when serving live traffic; more lightweight models, which allow serving thousands of users simultaneously, are necessary for deployment. However, the improved capability of the model is implicitly encoded in its learned weights, making it infeasible to transfer directly to smaller network architectures. A common alternative strategy is to use the learned representation (i.e., activations) of possibly larger models as a *guidance* for training smaller networks (Gou et al., 2021). From this perspective, *knowledge distillation* is the critical component in compressing larger models with better representations into more compact variants without sacrificing much of their quality.

Knowledge distillation refers to a set of techniques used for transferring information from typically a larger trained model, called the *teacher*, to a smaller model, called the *student* (Hinton et al., 2015; Anil et al., 2018; Xie et al., 2020). The goal of distillation is to improve the performance of

the student model by augmenting the knowledge of the teacher model with the raw information provided by the set of train examples. Since its introduction, several previous works have applied knowledge distillation to obtain improved results for language modeling (Sanh et al., 2019), image classification (Beyer et al., 2022), and robustness against adversarial attacks (Papernot et al., 2016).

The teacher's knowledge is typically encapsulated in the form of (expanded) soft labels, which are usually smoothened further by incorporating a temperature parameter at the output layer of the teacher model (Hinton et al., 2015; Müller et al., 2020). Other approaches consider matching the teacher's representations for a given input, typically in the penultimate layer, by the student (Romero et al., 2014). However, direct transfer of knowledge in the form of *compressed representations* has not been sufficiently explored before.

In this paper, we first introduce a technique for extracting compressed representations from an arbitrary layer of a trained model. Our technique conforms to the non-linearity of the layer's transfer function. Considering the geometry induced by the non-linearity allows us to learn more compact representations than the baseline approach of using standard PCA (Pearson, 1901). As an application to our layerwise representation learning approach, we explore the idea of directly transferring information from a teacher to a student in the form of *learned (fixed) principal directions* of arbitrary layers of the teacher model. Our focus for representation learning will be on a generalized form of the PCA method based on the broader class of Bregman divergences. The Bregman divergence (Bregman, 1967) induced by the strictly convex and differentiable function $F : \mathbb{R}^d \to \mathbb{R}$ between $\boldsymbol{u}, \boldsymbol{v} \in \operatorname{dom} F$ is defined as

$$D_F(\boldsymbol{u}, \boldsymbol{v}) = F(\boldsymbol{u}) - F(\boldsymbol{v}) - f(\boldsymbol{v}) \cdot (\boldsymbol{u} - \boldsymbol{v}), \tag{1}$$

where $f = \nabla F$ is the gradient function (commonly known as the *link* function). Bregman divergence is always non-negative and zero iff the arguments are equal. This broad class of divergences includes many well-known cases, such as the squared Euclidean and KL divergences as special cases. In addition, a Bregman divergence is not necessarily symmetric but satisfies a duality property in terms of the Bregman divergence between the dual variables. Let $F^*(\boldsymbol{u}^*) = \sup_{\boldsymbol{z}} \boldsymbol{z} \cdot \boldsymbol{u}^* - F(\boldsymbol{z})$ be the Legendre dual (Hiriart-Urruty & Lemaréchal, 2001) of $F$. Then, we can write $D_F(\boldsymbol{u}, \boldsymbol{v}) = D_{F^*}(\boldsymbol{v}^*, \boldsymbol{u}^*)$ where $\boldsymbol{u}^* = f(\boldsymbol{u})$ and $\boldsymbol{v}^* = f(\boldsymbol{v})$ are the pair of dual variables. When $F$ is strictly convex and differentiable, we have $f^* = f^{-1}$ and $\boldsymbol{u} = f^*(\boldsymbol{u}^*)$ and $\boldsymbol{v} = f^*(\boldsymbol{v}^*)$. Also, the derivative of a Bregman divergence with respect to the first argument takes the following simple form

$$\nabla_{\boldsymbol{u}} D_F(\boldsymbol{u}, \boldsymbol{v}) = f(\boldsymbol{u}) - f(\boldsymbol{v}). \tag{2}$$

The loss construction in Amid et al. (2022) provides a natural way of generating layerwise Bregman divergences for deep neural networks as line integrals of the strictly monotonic transfer functions. A dual view of such Bregman divergences in one dimension is illustrated in Figure 1(a) as areas under the curves of the strictly monotonic transfer function $f$ and its inverse. We will utilize such Bregman divergences for layerwise representation learning via an extension of the Bregman PCA algorithm. Note that the generalization of PCA to Bregman divergences was done by Collins et al. (2001) and extended in several previous works (Roy & Gordon, 2002; Acharyya, 2006; Chiquet et al., 2018). However, our generalization differs largely from the previous approaches by introducing a mean vector for handling non-centered data. Also, the orthogonality constraint in terms of Euclidean geometry is extended to orthonormal $k$-frames with respect to the Riemannian metric evaluated at the mean. To efficiently apply this constraint, we introduce a variant of the well-known QR

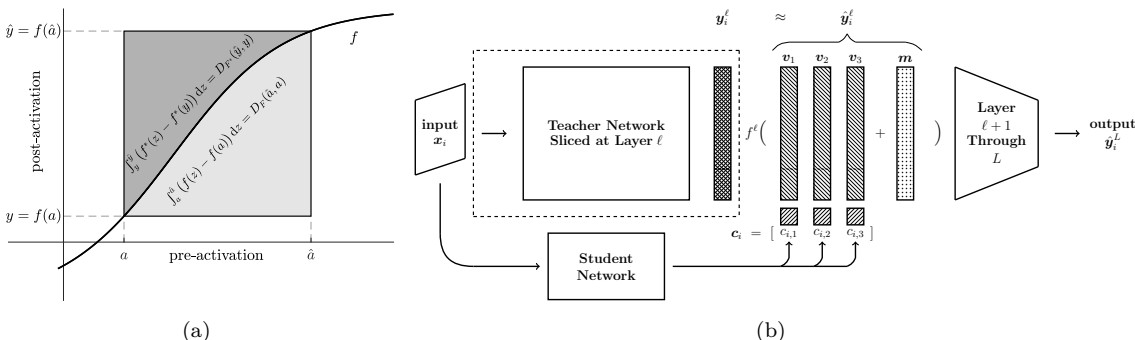

(a)  (b)

Figure 1: (a) A dual view of Bregman divergences as the area under the curves of the transfer function $f$ and its inverse $f^* = f^{-1}$ (Helmbold et al., 1999; Amid et al., 2022). (b) An illustration of knowledge distillation using Bregman representation learning. Given an input example $\boldsymbol{x}_i$, the output representation $\boldsymbol{y}_i^\ell$ of the teacher network at layer $\ell \in L$ is approximated as $\hat{\boldsymbol{y}}_i^\ell = f^\ell(\boldsymbol{V}\boldsymbol{c}_i + \boldsymbol{m})$ where the principal directions $\boldsymbol{V}$ and the mean vector $\boldsymbol{m}$ are learned using Bregman PCA on the full train set. The compression coefficients $\boldsymbol{c}_i$ are predicted by the student network, and the approximate representation $\hat{\boldsymbol{y}}_i^\ell$ is passed instead of $\boldsymbol{y}_i^\ell$ through the rest of the network from layer $\ell + 1$ to $L$. The teacher component inside the dashed bounding box can be discarded once the student network is trained.

decomposition. We apply our layerwise representation learning approach to distilling knowledge from a larger teacher network to a smaller student. Our extension of the Bregman PCA formulation can be viewed as adding an extra layer to the student network with fixed weights and biases. The distillation problem thus reduces to learning the corresponding compression coefficients of a given example by the student, which is passed as input to this layer. Our knowledge distillation approach of using Bregman representation learning is summarized in Figure 1(b).

## 1.1 Related Work

**Representation Learning and Knowledge Distillation:** There have been several lines of work that go beyond extracting knowledge from the model's predictive distribution. These include the work by Romero et al. (2014) where they add a regression loss between the representation of the teacher and a thinner (in the number of units) student representation. In their work, they learn a projection matrix to match the entire representation of the teacher rather than the relevant ones that are useful for the task. They also require the student to be thinner and deeper, which can result in difficulty in training. This line of work has been improved by Zagoruyko & Komodakis (2016) where the students are less deep and use the teacher's spatial attention map as guidance. Additionally, Czarnecki et al. (2017) propose incorporating the derivative of the teacher's prediction w.r.t to the input to be matched by the student, and this additional information has been shown to improve the distillation procedure. More recently, Tian et al. (2019) propose adding contrastive losses to transfer representation between two networks and show improvement over vanilla distillation. However, their procedure requires extra computation to push apart the teacher's representation of randomly sampled inputs from the student's representation of the actual inputs. Finally, Müller et al. (2020) tackle the task of extracting more information from the teacher model when the

number of classes is small. They propose creating sub-classes corresponding to every class, followed by soft-label training for distillation.

**Generalized PCA:** Collins et al. (2001) propose a generalization of the original PCA problem, defined for Gaussian models, to general exponential family distributions. Several extensions of this work focus on applying the framework to Poisson distributions (Chiquet et al., 2018) and compositional data (Avalos et al., 2018). Landgraf & Lee (2020b) apply logistic PCA for binary data for dimensionality reduction. Landgraf & Lee (2020a) also propose a majorization-minimization approach for solving the generalized PCA problem. Despite the success of earlier PCA extensions, a direct generalization of the original PCA formulation that includes a mean vector and a more principled orthogonality constraint on the directions was missing; our extension adds a mean vector while taking the local geometry of the space around the mean into account.

## 2 Extended Bregman PCA

Principal Component Analysis (PCA) (Pearson, 1901; Shlens, 2014) is perhaps one of the most commonly used techniques for data compression, dimensionality reduction, and representation learning. In the simplest form, the PCA problem is defined as minimizing the *compression loss* of representing a set of points as linear combinations of a set of orthonormal *principal directions*. More concretely, given $\mathcal{X} = \{\boldsymbol{x}_i \in \mathbb{R}^d\}$ and $k < d$, the PCA problem can be formulated as finding the *mean vector* $\boldsymbol{m} \in \mathbb{R}^d$ and principal directions $\boldsymbol{V} \in \mathbb{R}^{d \times k}$ where $\boldsymbol{V}^\top \boldsymbol{V} = \boldsymbol{I}_k$ such that the *compression loss*,

$$\boldsymbol{m}, \boldsymbol{V}, \{\boldsymbol{c}_i\} = \underset{\left\{ \substack{\tilde{\boldsymbol{m}} \in \mathbb{R}^d, \widetilde{\boldsymbol{V}} \in \mathrm{St}_{d,k}, \\ \{\tilde{\boldsymbol{c}}_i \in \mathbb{R}^k\}} \right\}}{\arg\min} \sum_i \|\boldsymbol{x}_i - (\tilde{\boldsymbol{m}} + \widetilde{\boldsymbol{V}} \tilde{\boldsymbol{c}}_i)\|^2, \tag{3}$$

is minimized. Here, $\mathrm{St}_{d,k} = \{\boldsymbol{U} \in \mathbb{R}^{d \times k} : \boldsymbol{U}^\top \boldsymbol{U} = \boldsymbol{I}_k\}$ denotes the Stiefel manifold of $k$-frames in $\mathbb{R}^d$ (Hatcher, 2000) and $\boldsymbol{c}_i \in \mathbb{R}^k$ corresponds to the *compression coefficients* of $\boldsymbol{x}_i$. The problem in Eq. (3) can be solved effectively in two steps. First, we note that $\boldsymbol{m}$ can be viewed as a constant shared representation for all points in $\mathcal{X}$ (for which the code length is zero) that minimizes the total compression loss. With this interpretation, the mean vector can be written as the minimizer of

$$\boldsymbol{m} = \underset{\tilde{\boldsymbol{m}} \in \mathbb{R}^d}{\arg\min} \sum_i \|\boldsymbol{x}_i - \tilde{\boldsymbol{m}}\|^2, \tag{4}$$

for which, the solution corresponds to the geometric mean $\boldsymbol{m} = \frac{1}{|\mathcal{X}|} \sum_i \boldsymbol{x}_i$. By fixing $\boldsymbol{m}$, the solution for $\boldsymbol{V}$ and $\{\boldsymbol{c}_i\}$ can be obtained by enforcing the orthonormality constraints using a set of Lagrange multipliers and setting the derivatives to zero. The solution to $\boldsymbol{V}$ amounts to the top-$k$ eigenvectors of the covariance matrix $1/|\mathcal{X}| \sum_i (\boldsymbol{x}_i - \boldsymbol{m})(\boldsymbol{x}_i - \boldsymbol{m})^\top$ and $\boldsymbol{c}_i = \boldsymbol{V}^\top (\boldsymbol{x}_i - \boldsymbol{m})$ corresponds to the projection of the centered point onto the column space of $\boldsymbol{V}$. Noticeably, the online variants of PCA, such as Oja's algorithm (Oja, 1982), alternatively apply a gradient step on $\boldsymbol{V}$ and project the update onto $\mathrm{St}_{d,k}$ by an application of QR decomposition (Golub & Van Loan, 1996).

Given a strictly convex and differentiable function $F : \mathbb{R}^d \to \mathbb{R}$ with link function $f = \nabla F$, we cast the generalized Bregman PCA problem as approximating $\boldsymbol{x}_i \in \mathcal{X}$ as a linear combination of a set of orthonormal principal directions in the dual space. This formulation reduces to Eq. (3) for the choice of $f = \mathrm{id}_d$, which corresponds to the squared Euclidean divergence. However, before defining the objective function formally, we consider the problem of finding a generalized mean in the dual

space as follows. Let $\mathcal{X} = \{\boldsymbol{x}_i \in \mathrm{dom}\, F^*\}$ be a set of given data points. We define the generalized mean vector $\boldsymbol{m}$ as the minimizer of the following objective,

$$\boldsymbol{m} = \underset{\tilde{\boldsymbol{m}} \in \mathrm{dom}\, F}{\arg\min} \sum_i D_{F^*}(\boldsymbol{x}_i, f(\tilde{\boldsymbol{m}}))\,. \tag{5}$$

Note that the above Eq. (5) is a direct generalization of Eq. (4) in terms of finding a shared constant representation for all points in $\mathcal{X}$ that minimizes a notion of *Bregman compression loss*. The following proposition states the solution of the generalized mean in a closed form.

**Proposition 1.** *The generalized dual mean in Eq. (5) can be written as*

$$\boldsymbol{m} = f^*\big(\frac{1}{|\mathcal{X}|} \sum_i \boldsymbol{x}_i\big)\,. \tag{6}$$

Thus, the dual mean simply corresponds to the dual of the arithmetic mean of the data points. When $f = \mathrm{id}_d$, i.e., the identity function, the dual mean reduces to the arithmetic mean.

Given the definition of the dual mean in Eq. (6), we now extend the vanilla PCA formulation in Eq. (3) to the class of Bregman divergences. First, we note that the geometry of the space of principal directions is altered when switching from the squared loss to a more general Bregman divergence. Specifically, given the convex function $F$ of a Bregman divergence, the inner product is locally governed by the Riemannian metric (Lee, 2006),

$$D_F(\boldsymbol{u} + \delta\boldsymbol{u}, \boldsymbol{u}) = D_F(\boldsymbol{u}, \boldsymbol{u} + \delta\boldsymbol{u}) \approx \tfrac{1}{2}\, \delta\boldsymbol{u}^\top \boldsymbol{H}_F(\boldsymbol{u})\delta\boldsymbol{u}\,,$$

where $\delta\boldsymbol{u}$ is a small perturbation and $\boldsymbol{H}_F = \nabla^2 F$ is the *Hessian* of $F$. Thus, the definition of orthonormality needs to conform to the new geometry imposed by the Bregman divergence. In the following, we extend the definition of a Stiefel manifold to include a Riemannian metric. Recall that for a strictly convex function $F$, we have $\boldsymbol{H}_F \in \mathbb{S}_{++}^n$ where $\mathbb{S}_{++}^n$ denotes the set of $n \times n$ symmetric positive definite matrices. We now formally define the Riemann-Stiefel manifold.

**Definition 1.** *The Riemann-Stiefel manifold of $k$-frames in $\mathbb{R}^d$ with respect to the Riemannian metric $\boldsymbol{M} = \boldsymbol{M}(\boldsymbol{v}) \in \mathbb{S}_+^d, \boldsymbol{v} \in \mathbb{R}^d$ is defined as*

$$\mathrm{St}_{d,k}^{(\boldsymbol{M})} = \{\boldsymbol{U} \in \mathbb{R}^{d \times k} : \boldsymbol{U}^\top \boldsymbol{M} \boldsymbol{U} = \boldsymbol{I}_k\}\,.$$

Note that for the Euclidean geometry, $\boldsymbol{M}(\boldsymbol{v}) = \boldsymbol{I}_d$ for all $\boldsymbol{v} \in \mathbb{R}^d$. Thus, the definition of $\mathrm{St}_{d,k}^{(\boldsymbol{M})}$ recovers $\mathrm{St}_{d,k}^{(\boldsymbol{I}_d)} = \mathrm{St}_{d,k}$ and we arrive at the orthonormality in the sense of the Euclidean geometry.

We now formulate the generalized Bregman PCA problem following the local geometry of the strictly convex function at $\boldsymbol{m}$. Let $\mathcal{X} = \{\boldsymbol{x}_i \in \mathrm{dom}\, F^*\}$ be a set of given points. We define the generalized Bregman PCA as finding a linear combination of a set of generalized principal directions that minimizes the Bregman compression loss:

$$\boldsymbol{V}, \{\boldsymbol{c}_i\} = \underset{\left\{\substack{\widetilde{\boldsymbol{V}} \in \mathrm{St}_{d,k}^{(\boldsymbol{H}_F(\boldsymbol{m}))}, \\ \{\tilde{\boldsymbol{c}}_i \in \mathbb{R}^k\}}\right\}}{\arg\min} \sum_i D_{F^*}(\boldsymbol{x}_i, f(\boldsymbol{m} + \widetilde{\boldsymbol{V}}\tilde{\boldsymbol{c}}_i))\,, \tag{7}$$

where $\boldsymbol{m}$ is given via Eq. (6) and $\boldsymbol{H}_F(\boldsymbol{m}) = \nabla^2 F(\boldsymbol{m})$. Note that the constraint $\boldsymbol{V} \in \mathrm{St}_{d,k}$ in Eq. (3) is now replaced with $\boldsymbol{V} \in \mathrm{St}_{d,k}^{(\boldsymbol{H}_F(\boldsymbol{m}))}$, i.e., using the Riemannian metric induced by the Hessian of the convex function $F$ evaluated at the dual mean $\boldsymbol{m}$ (Amari, 2016).

## 2.1 Optimization

The generalized Bregman PCA objective in Eq. (7) does not yield a closed-form solution in terms of $\boldsymbol{V}$ and $\{\boldsymbol{c}_i\}$. However, the problem is convex in both $\boldsymbol{V}$ and $\{\boldsymbol{c}_i\}$ and can be solved iteratively by applications of gradient descent steps. Let $\hat{\boldsymbol{x}}_i := f(\boldsymbol{m} + \boldsymbol{V}\boldsymbol{c}_i)$ denote the approximation of $\boldsymbol{x}_i$. For the compression coefficients $\{\boldsymbol{c}_i\}$, we apply

$$\boldsymbol{c}_i^{\text{new}} = \boldsymbol{c}_i - \eta_a \boldsymbol{V}^\top (\hat{\boldsymbol{x}}_i - \boldsymbol{x}_i), \tag{8}$$

where $\eta_a > 0$ denotes the learning rate. Updating $\boldsymbol{V}$ involves two steps: gradient updates followed by a projection onto $\text{St}_{d,k}^{(\boldsymbol{H}_F(\boldsymbol{m}))}$. We apply gradient descent updates,

$$\boldsymbol{V}^{\text{new}} = \boldsymbol{V} - \eta_V \sum_i (\hat{\boldsymbol{x}}_i - \boldsymbol{x}_i)\,\boldsymbol{c}_i^\top, \tag{9}$$

where $\eta_V > 0$ denotes the learning rate. The next step involves projecting $\boldsymbol{V}^{\text{new}}$ back to $\text{St}_{d,k}^{(\boldsymbol{H}_F(\boldsymbol{m}))}$. As we shall see, this projection needs to be applied only once at the end of optimization since both $\boldsymbol{V}$ and $\{\boldsymbol{c}_i\}$ are trained using gradient descent (any intermediate factor can be absorbed into the gradients). For the vanilla PCA where $\boldsymbol{H}_F(\boldsymbol{m}) = \boldsymbol{I}_d$, this can be applied easily by an application of QR decomposition. We provide a simple modification of the standard QR decomposition algorithm that achieves this for any $\boldsymbol{H}_F(\boldsymbol{m}) \in \mathbb{S}_{++}^n$, with almost no additional overhead in practice for our application.

## 2.2 QR Decomposition on the Riemann-Stiefel Manifold

A QR decomposition is a factorization of a matrix $\boldsymbol{A} \in \mathbb{R}^{m \times n}$ where $n \leq m$ into a product $\boldsymbol{A} = \boldsymbol{Q}\boldsymbol{R}$ where $\boldsymbol{Q} \in \text{St}_{\{m,n\}}$ and $\boldsymbol{R} \in \mathbb{R}^{n \times n}$ is an upper-triangular matrix. The first factor $\boldsymbol{Q}$ can be viewed as an orthonormalization of columns of $\boldsymbol{A}$, similar to the result of a Gram-Schmidt procedure. However, QR decomposition provides a more numerically stable procedure in general. The method of Householder reflections (Golub & Van Loan, 1996) is the most common algorithm for QR decomposition.

---
**Algorithm 1** `RS-QR`$(\boldsymbol{A}, \boldsymbol{M})$: QR Decomposition on the Riemann-Stiefel Manifold

---
    **Input:** matrix $\boldsymbol{A} \in \mathbb{R}^{m \times n}$ s.t. $n \leq m$, Riemannian metric $\boldsymbol{M} \in \mathbb{S}_+^m$
    **Output:** $\boldsymbol{Q} \in \mathbb{R}^{m \times n}$ and $\boldsymbol{R} \in \mathbb{R}^{n \times n}$ factors such that $\boldsymbol{A} = \boldsymbol{Q}\boldsymbol{R}$ and $\boldsymbol{Q}^\top \boldsymbol{M} \boldsymbol{Q} = \boldsymbol{I}_n$
    $\widetilde{\boldsymbol{Q}}, \boldsymbol{R} \leftarrow \texttt{QR}(\sqrt{\boldsymbol{M}}\boldsymbol{A})$
    $\boldsymbol{Q} \leftarrow \sqrt{\boldsymbol{M}^{-1}}\widetilde{\boldsymbol{Q}}$
    **Return:** $\boldsymbol{Q}, \boldsymbol{R}$

---

The following theorem provides a procedure that extends the standard QR decomposition to produce conjugate factors $\boldsymbol{Q}^\top \boldsymbol{M} \boldsymbol{Q} = \boldsymbol{I}_n$ for a given $\boldsymbol{M} \in \mathbb{S}_{++}^n$.

**Theorem 1.** *Let* QR *denote the procedure that returns the QR factors. Given* $\boldsymbol{M} \in \mathbb{S}_{++}^n$ *and* $\boldsymbol{A} \in \mathbb{R}^{m \times n}$, *let* $\widetilde{\boldsymbol{Q}}, \boldsymbol{R} = \texttt{QR}(\sqrt{\boldsymbol{M}}\boldsymbol{A})$. *Then, the matrix* $\boldsymbol{Q} = \sqrt{\boldsymbol{M}^{-1}}\widetilde{\boldsymbol{Q}}$ *corresponds to the first factor of the QR decomposition of* $\boldsymbol{A}$ *on the Riemann-Stiefel manifold such that* $\boldsymbol{A} = \boldsymbol{Q}\boldsymbol{R}$ *and* $\boldsymbol{Q}^\top \boldsymbol{M} \boldsymbol{Q} = \boldsymbol{I}_m$.

The QR decomposition on the Riemann-Stiefel manifold imposes almost no extra overhead compared to standard QR when the matrix $\boldsymbol{M}$ is diagonal. As we shall see, this is in fact the case for the local metric induced by the majority of the commonly used elementwise transfer functions such as leaky ReLU, sigmoid, and tanh. We defer the special case of the QR decomposition for the

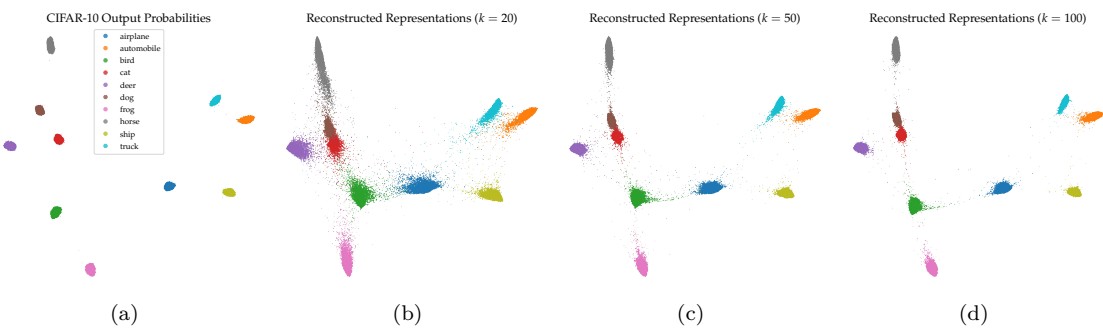

Figure 2: Visualization of (a) the output probabilities of the teacher model $\{\boldsymbol{y}_i^L\}$ and (b)-(d) the reconstructed representation of the training examples at the penultimate leaky ReLU layer $\{\hat{\boldsymbol{y}}_i^{L-1}\}$ using TriMap (Amid & Warmuth, 2019). As the number of components $k = 20, 50, 100$ increases, the approximated representations become better separable. Interestingly, TriMap reveals that the representations at layer $L-1$ form similar clusters as the ones in layer $L$ of the original teacher model in (a).

---

**Algorithm 2** Bregman PCA with Mean

---

**Input:** $\mathcal{X} = \{\boldsymbol{x}_i \in \operatorname{dom} F^* \subseteq \mathbb{R}^d\}$, number of components $k < d$
**Output:** dual mean $\boldsymbol{m} \in \operatorname{dom} F$, $\boldsymbol{V} \in \operatorname{St}_{d,k}^{(\boldsymbol{H}_F(\boldsymbol{m}))}$, $\{\boldsymbol{c}_i \in \mathbb{R}^k\}$
$\boldsymbol{m} \leftarrow f^*\left(\frac{1}{|\mathcal{X}|} \sum_i \boldsymbol{x}_i\right)$
initialize $\boldsymbol{V}$ and $\{\boldsymbol{c}_i\}$
**repeat**
    **for** $i \in [|\mathcal{X}|]$ **do**
        $\hat{\boldsymbol{x}}_i \leftarrow f(\boldsymbol{m} + \boldsymbol{V}\boldsymbol{c}_i)$
        $\boldsymbol{c}_i \leftarrow \boldsymbol{c}_i - \eta_a \boldsymbol{V}^\top(\hat{\boldsymbol{x}}_i - \boldsymbol{x}_i)$
    $\boldsymbol{V} \leftarrow \boldsymbol{V} - \eta_V \sum_i (\hat{\boldsymbol{x}}_i - \boldsymbol{x}_i)\boldsymbol{c}_i^\top$
**until** $\boldsymbol{V}, \{\boldsymbol{c}_i\}$ not converged
$\boldsymbol{V}, \boldsymbol{T} \leftarrow \text{RS-QR}(\boldsymbol{V}, \boldsymbol{H}_F(\boldsymbol{m}))$
**return** $\boldsymbol{m}, \boldsymbol{V}, \{\boldsymbol{T}\boldsymbol{c}_i\}$

---

softmax function to Appendix B. Our generalized Bregman PCA algorithm with mean is given in Algorithm 2. We omit the case of the softmax function in the main algorithm for simplicity (see Appendix C for the full algorithm).

## 3 Representation Learning of Deep Neural Networks

One important application of our Bregman PCA is learning the representations of a deep neural network in each layer. Specifically, in a deep neural network, each layer transforms the representation that receives from the previous layer and passes it to the next layer. In a given layer, we are interested in learning the mean and principal directions that can encapsulate the representations of all training examples in that layer. Although the vanilla PCA might be the naïve choice for this purpose, we will consider learning more effective representations using our extended Bregman PCA approach.

A natural choice of a Bregman divergence for a layer having a strictly increasing transfer function is the one induced by the convex integral function of the transfer function (Amid et al., 2022). In Bregman PCA, we essentially want to minimize the *matching loss* of the transfer function $f$ instead of the quadratic compression loss used for the vanilla PCA. The matching loss was introduced by Helmbold et al. (1999); Kivinen & Warmuth (2001) and is also the main tool in the recent work on training deep neural networks (Amid et al., 2019; 2022). Specifically, let $\boldsymbol{a}_i^{(\ell)} \in \mathbb{R}^d$ and $\boldsymbol{y}_i^{(\ell)} \in \mathbb{R}^d$ respectively be the pre and post (transfer function) activations of a neural network for a given input example $\boldsymbol{x}_i$ at layer $\ell \in [L]$. The layer applies an (elementwise) strictly increasing transfer function $f^{(\ell)}$, i.e., $\boldsymbol{y}_i^{(\ell)} = f^{(\ell)}(\boldsymbol{a}_i^{(\ell)})$. Let $F^{(\ell)}$ denote a convex integral function of $f^{(\ell)}$, i.e., $f^{(\ell)} = \nabla F^{(\ell)}$. Then, for a given set of input examples $\mathcal{X} = \{\boldsymbol{x}_i\}$ having post-activation representations $\mathcal{Y}^{(\ell)} = \{\boldsymbol{y}_i^{(\ell)}\}$, we can cast the Bregman PCA problem in layer $\ell \in [L]$ as learning $\boldsymbol{V}^{(\ell)} \in \mathrm{St}_{d,k}^{(\boldsymbol{H}_{F^{(\ell)}}(\boldsymbol{m}^{(\ell)}))}$ and $\{\boldsymbol{c}_i^{(\ell)}\}$ with $\boldsymbol{m}^{(\ell)} = f^{*(\ell)}\big(\frac{1}{|\mathcal{Y}^{(\ell)}|}\sum_i \boldsymbol{y}_i^{(\ell)}\big)$ that minimize the objective

$$\sum_i D_{F^{*(\ell)}}(\boldsymbol{y}_i^{(\ell)}, f^\ell(\boldsymbol{m}^{(\ell)} + \boldsymbol{V}^{(\ell)}\boldsymbol{c}_i^{(\ell)})). \tag{10}$$

We defer the technical details of constructing the matching loss of a transfer function for a given layer to Appendix D. We explore several ideas based on this layerwise construction, including learning the principal directions in each layer in the next section. Specifically, we consider representation learning in the final softmax layer as well as the fully-connected (penultimate) leaky ReLU layer before the final softmax layer of a ResNet-18 model. Nonetheless, our construction is applicable to other types of layers, such as convolutions. We then consider knowledge distillation using the representations learned from the ResNet-18 teacher model to train a smaller convolutional student model on the same dataset. Our knowledge distillation approach is illustrated in Figure 1. Our approach consists of learning the dual mean $\boldsymbol{m}$ and the principal directions $\boldsymbol{V}$ in the $\ell$-th layer of the teacher network. The representation $\boldsymbol{y}_i^\ell$ of a training example $\boldsymbol{x}_i$ is then approximated by $\hat{\boldsymbol{y}}_i^\ell = f^\ell(\boldsymbol{V}\boldsymbol{c}_i + \boldsymbol{m})$ where the compression coefficients $\boldsymbol{c}_i \in \mathbb{R}^k$ are predicted by a smaller student network. The approximated representation $\hat{\boldsymbol{y}}_i^\ell$ can then be passed through the rest of the pre-trained teacher network or a smaller network that is trained from scratch to predict the output labels. This approach can easily be extended to distilling information from several different layers of the teacher model. However, we defer exploring such extensions to future work and focus on knowledge distillation using the representation of a single layer of the teacher model.

## 4 Experiments

We conduct the first set of experiments on the CIFAR-10 and CIFAR-100 datasets, each consisting of 50,000 train and 10,000 test images of size $32 \times 32$ from, respectively, 10 and 100 classes.[1] In order to extract representations, we first train a PreAct ResNet-18 model using SGD with a Nesterov momentum optimizer and a batch size of 128. The only modification we apply to the network is replacing the global average pooling layer before the final dense layer with a flattening operator. This modification yields a representation of dimension 8192 before the final layer. We also change the transfer function of that layer from a ReLU to a leaky ReLU with a small negative slope of $\beta = 1e-4$ to obtain a strictly monotonic transfer function. The network achieves 92.62% and 70.44% top-1 test accuracy on CIFAR-10 and CIFAR-100, respectively. We then pass the train

---

[1]Available at `https://www.cs.toronto.edu/~kriz/cifar.html`.

and test examples of both datasets and store the leaky ReLU post-activations ($d = 8192$) and the output softmax probabilities ($d = 10$ and $d = 100$ for CIFAR-10 and CIFAR-100, respectively) of each example. In the following, we consider several experiments for compressing each of these representations. We use normalized SGD with an exponential moving average (EMA) momentum for optimizing our Bregman PCA problem. At last, we also provide results on the ImageNet-1k datasets (Deng et al., 2009). Further details about the experiments are provided in Appendix E.

### 4.1 PCA on Output Probabilities

We first consider the problem of compressing the output probabilities. This problem corresponds to a Bregman PCA with a softmax link function and a KL divergence. For comparison, we consider the vanilla PCA (Eq. (3)) in the pre-activations (i.e., logits) domain. Note that PCA is not directly applicable to the post-activation probabilities, as the reconstructed examples may not correspond to valid probability distributions. We also consider CoDA PCA (Avalos et al., 2018) by augmenting a constant bias of 1 to the coefficient to handle the non-centered case, as suggested in the paper. We use SGD with a heavy-ball momentum for solving the CoDA PCA problem. We report the KL divergence between the model probabilities and the reconstructed probabilities for the train and test sets as the performance measure.

The average KL divergence values between the original output and the reconstructed probabilities are shown in Figure 3(a)-(b). As can be seen from the figure, Bregman PCA yields a much lower loss compared to vanilla PCA and CoDA PCA. Also, notice that CoDA-PCA becomes inefficient as the dimension of the problem ($d$ or $k$) increases.

### 4.2 PCA on Leaky ReLU Outputs

We consider the problem of compressing the representation in the leaky ReLU layer before the final dense layer (i.e., the penultimate layer). The convex conjugate function $F_\beta^*$ corresponding to a leaky ReLU transfer function $f_\beta(\boldsymbol{u}) = \max(\boldsymbol{u}, \boldsymbol{0}) - \beta \max(-\boldsymbol{u}, \boldsymbol{0})$ with a slope $\beta > 0$ amounts to $F_\beta^*(\boldsymbol{u}) = {}^1\!/_2\, \boldsymbol{u} \odot f_{\beta^{-1}}(\boldsymbol{u})$ where $\odot$ denotes Hadamard product. For this problem, we consider the vanilla PCA on the pre and post-leaky ReLU activations as well as our Bregman PCA. In order to compare the methods, we pass the reconstructed representation of each example through the final layer of the trained ResNet-18 model and measure the top-1 accuracy.

The results are given in Figure 3(c)-(d). Our Bregman PCA works significantly better than vanilla PCA for reconstructing the representation and yields a much higher classification accuracy at the output. In order to visualize these representations, we illustrate the 2-D TriMap (Amid & Warmuth, 2019) projections of the CIFAR-10 output probabilities as well as the reconstructed representations of the examples at the leaky ReLU layer using our method in Figure 3. We can see from the figure that the two representations, although one layer apart, form very similar clusters. Also, as the number of components $k$ increases, the leaky ReLU representations become better separable.

### 4.3 Distillation Results on CIFAR-10/100

We conduct knowledge distillation experiments using our Bregman Representation Learning (**BRL**) method. The architecture we consider for the experiments is shown in Figure 1(b). The student network consists of a small convolutional neural network with 5 convolutional layers of size $[128\,(\times 2),$

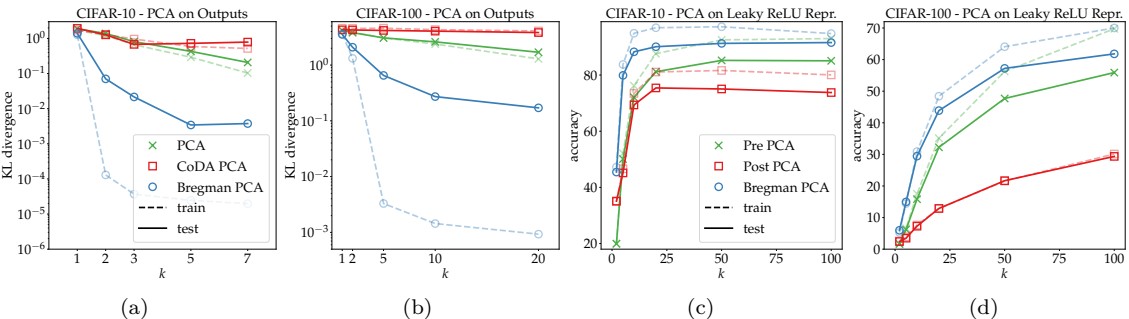

Figure 3: (a) and (b): KL divergence between the original and reconstructed class probabilities of the train and test examples at the final layer. (Lower values are better.) Our Bregman PCA approach significantly outperforms both vanilla PCA on the logits as well as CoDA PCA. (c) and (d): Top-1 classification accuracy of the reconstructed leaky ReLU representations when passed through the rest of the network. (Higher values are better.) Our Bregman PCA works significantly better than vanilla PCA for reconstructing the representation and yields a much higher classification accuracy.

$256\,(\times 2)$, 512] followed by a dense linear layer which outputs $k$ coefficients. These coefficients are then passed through a dense layer (with weights $V$ and bias $m$), which outputs a representation of size 8192. The final dense (readout) layer with softmax transfer function converts this representation into output class probabilities.

For comparison, we consider the following approaches: 1) Baseline model for which all the weights are randomly initialized and are trained using the same training examples as the teacher, 2) Baseline + Readout where all the weights are initialized randomly except the readout layer, which is set to the pre-trained weights of the teacher, 3) Baseline Distillation where we use a convex combination of the teacher's soft labels (for which we also tune a temperature for the logits) with the one-hot training labels. The ratio of the teacher's soft labels to train labels is tuned for each case. 4) BRL, where we randomly initialize all the weights except $V$ and $m$, which are set to the values obtained by applying our Bregman PCA on the leaky ReLU representations of the training examples, as in Section 4.2. 5) BRL + Readout, which is similar to BRL, but we set the readout layer weights to the teacher weights. Each model is trained for 51 epochs using a batch size of 128 using SGD with a Nesterov momentum optimizer with a linear decay schedule. In each case, we tune the learning rate and momentum hyper-parameters.

To train BRL, we first train the student model by minimizing a squared loss between the predicted coefficients of the student and the compression coefficients obtained by directly applying our Bregman PCA to the teacher's representations (as in the previous section). This essentially reduces the training of the student network from classification to a regression problem. We find that this pre-training on the teacher coefficients significantly improves the convergence of the student model. After training on the compression coefficients for 25 epochs, we switch to directly minimizing the cross-entropy loss at the output layer for the rest of the iterations.

To disentangle the improvements due to better information transfer for knowledge distillation from the gains obtained due to data augmentation, we conduct the experiments on the original examples without any additional augmentations (such as random horizontal flip, random crop, Mixup (Zhang et al., 2018), etc.), which are standard for training these models. This resembles a limited sample

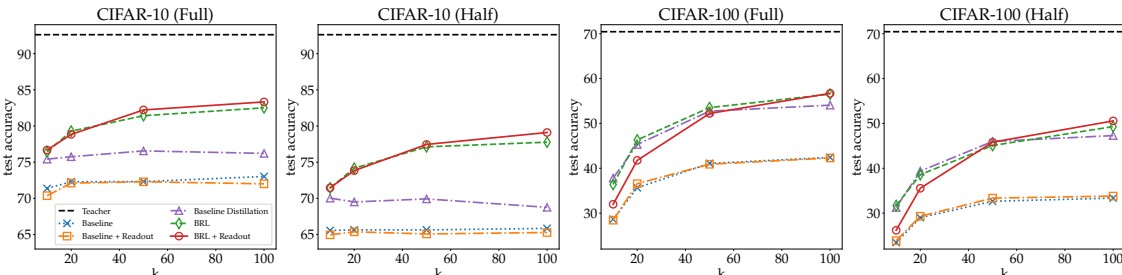

Figure 4: Results of training different baselines, including soft-label teacher-student distillation and our BRL distillation approach on the full and 50% randomly subsampled (half) CIFAR-10 and CIFAR-100 datasets. In each case, + Readout indicates using the teacher's pre-trained readout weights for the student.

regime where the number of training examples is relatively small. We conduct additional experiments by incorporating these augmentations and observe improvements for all baselines (see Appendix F). In order to highlight the limited samples aspect even further, we also repeat each experiment using only half of the training dataset, which is uniformly sampled from the full dataset (and fixed for all methods).

The results of training the smaller student model are shown in Figure 4. We observe a significant improvement in accuracy using our BRL distillation approach across different numbers of components ($k$). For CIFAR-10, even with a small $k$, BRL shows a major advantage over soft-label teacher-student training. For CIFAR-100, these improvements are minor for smaller $k$ values as the leaky ReLU representations require a relatively larger number of components to be reconstructed efficiently compared to CIFAR-10 (see Figure 3(c)-(d)). However, as $k$ increases, we observe that the added benefit from more Bregman representations outweighs the benefit of a higher capacity model (due to larger $V$) for the baseline distillation approach using soft labels (56.5% top-1 accuracy for BRL compared to 54.1% for Baseline Distillation).

Additionally, in Figure 4, we observe that the gap between BRL and soft-label distillation becomes even more prominent as the number of training examples decreases (6.3% gap for full vs. 9.0% gap for half CIFAR-10 dataset). This shows that BRL is a more sample-efficient approach for transferring information between the teacher and the student models. Finally, in both cases, we see a marginal improvement for using the pre-trained readout layer weights over training the readout layer from scratch for larger $k$ values.

### 4.4 Distillation Results on ImageNet-1k

We consider the problem of distilling a PreAct ResNet-50 teacher model trained on the ImageNet-1k dataset into a PreAct ResNet-18 student. Similar to the previous experiments, we replace the transfer function of the penultimate layer with a leaky ReLU with a negative slope of $\beta = 1e-2$. We train all models on an $8 \times 8$ TPU v3 hardware. The teacher model is trained for 90 epochs using SGD with momentum with a piecewise constant schedule and achieves a 76.2% top-1 test accuracy. The baseline ResNet-18 model using the same training procedure achieves a 70.1% top-1 test accuracy. As the distillation baseline, we consider soft-label teacher-student training (Hinton et al., 2015) and tune the soft-label ratio and the logit temperature, and train the model for 90

Table 1: ImageNet-1k distillation results using a ResNet-50 teacher and a ResNet-18 student. Each model is trained for 90 epochs. We also report the number of parameters of each model and the runtime on the same $8 \times 8$ TPU v3 hardware.

| Model | Top-1 Accuracy | Top-5 Accuracy | #Params | Runtime |
|---|---|---|---|---|
| ResNet-50 Teacher | 76.2% | 93.0% | 25.6M | 1.7h |
| ResNet-18 Baseline | 70.1% | 89.4% | 11.7M | 1.3h |
| ResNet-18 Soft-label Distillation | 71.5% | 90.4% | 11.7M | 2.4h |
| ResNet-18 BRL (Ours) | **72.6**% | **91.1**% | 15.6M | 3.9h |

epochs using the same optimization hyperparameters as the ResNet-18 baseline. The soft-label distillation baseline achieves a 71.5% top-1 test accuracy.

The teacher's penultimate layer has dimension $d = 2048$, which we compress into $k = 512$ principal directions. To extract the Bregman representations and the mean vector for BRL, we simply apply our Bregman PCA algorithm in an online fashion in the last 10 epochs of the teacher model's training. (To find the mean, we apply an EMA.) We then import these representations into the student model and keep them fixed. We replace the last ResNet-18 layer with a linear layer to produce the 512 compression coefficients. We also use the weights of the teacher's readout layer at the student's output. We replace the optimizer with Adam (Kingma & Ba, 2014) with a cosine learning rate schedule. Similar to soft-label distillation, we consider the teacher's soft labels with the same ratio as the distillation baseline. We also add a squared regularizer term between the teacher's target and the student's predicted compression coefficients. Our BRL method combined with the teacher's soft labels achieves a 72.6% top-1 test accuracy after 90 epochs. The results are summarized in Table 1. We also report the number of parameters of each model and the runtime on the same hardware. In summary, our BRL distillation significantly improves the test accuracy while introducing a slight overhead in terms of the number of parameters and the runtime.

## 5 Conclusion and Future Work

We presented a new direction for knowledge distillation based on directly transferring the compressed Bregman representations of the teacher network into a student network. Our work significantly advances the previous approaches for knowledge distillation which rely on either training using soft teacher labels or matching the representations of the teacher and the student in the intermediate layers. Our construction is more flexible and can be extended to training a cascade of student networks. Specifically, a teacher network can be split into non-overlapping sub-networks where each portion is approximated by a smaller student. However, other than the first student, each student receives the output of the previous student as its input, thus, forming a *cascade* of networks to downsize the teacher model. Such training strategies are interesting future directions for our technique.

**Limitations and Broader Impact** One limitation of our current approach is that it only applies to layers with strictly monotonic transfer functions. The extension of our work to non-monotonic transfer functions is an interesting open problem. In addition, our layerwise representation learning improves the quality of the distilled models. Reducing the size of the model while maintaining its quality decreases the power usage of the model at the serving time. However, our method requires a longer training time compared to standard soft-label distillation. Such overhead is only reasonable if the model is utilized for inference for a reasonably long period of time.

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

## A  Omitted Proofs

**Proposition 1.** *The generalized dual mean in Eq. (5) can be written as*

$$\boldsymbol{m} = f^*\big(\frac{1}{|\mathcal{X}|}\sum_i \boldsymbol{x}_i\big)\,. \tag{6}$$

*Proof.* Using the duality property of Bregman divergences, Eq. (5) can be written as

$$\min_{\tilde{\boldsymbol{m}}\in\mathrm{dom}\,F}\sum_i D_{F^*}(\boldsymbol{x}_i, f(\tilde{\boldsymbol{m}})) = \min_{\tilde{\boldsymbol{m}}\in\mathrm{dom}\,F}\sum_i D_F(\tilde{\boldsymbol{m}}, f^*(\boldsymbol{x}_i))\,.$$

Applying Eq. (2) and setting the derivative of above with respect to $\tilde{\boldsymbol{m}}$ to zero, we have

$$\sum_i \big(f(\boldsymbol{m}) - f(f^*(\boldsymbol{x}_i))\big) = |\mathcal{X}|\cdot f(\boldsymbol{m}) - \sum_i \boldsymbol{x}_i = \boldsymbol{0}\,.$$

Rearranging the term and applying the inverse yields the result. $\qquad\square$

**Theorem 1.** *Let* QR *denote the procedure that returns the QR factors. Given $\boldsymbol{M}\in\mathbb{S}^n_{++}$ and $\boldsymbol{A}\in\mathbb{R}^{m\times n}$, let $\widetilde{\boldsymbol{Q}}, \boldsymbol{R} = $ QR$(\sqrt{\boldsymbol{M}}\boldsymbol{A})$. Then, the matrix $\boldsymbol{Q} = \sqrt{\boldsymbol{M}^{-1}}\widetilde{\boldsymbol{Q}}$ corresponds to the first factor of the QR decomposition of $\boldsymbol{A}$ on the Riemann-Stiefel manifold such that $\boldsymbol{A} = \boldsymbol{Q}\boldsymbol{R}$ and $\boldsymbol{Q}^\top\boldsymbol{M}\boldsymbol{Q} = \boldsymbol{I}_m$.*

*Proof.* Since $\widetilde{\boldsymbol{Q}}\boldsymbol{R} = \sqrt{\boldsymbol{M}}\boldsymbol{A}$, we have

$$\boldsymbol{Q}\boldsymbol{R} = \sqrt{\boldsymbol{M}^{-1}}\widetilde{\boldsymbol{Q}}\boldsymbol{R} = \sqrt{\boldsymbol{M}^{-1}}\sqrt{\boldsymbol{M}}\boldsymbol{A} = \boldsymbol{A}\,,$$

and $\boldsymbol{Q}$ satisfies

$$\boldsymbol{Q}^\top\boldsymbol{M}\boldsymbol{Q} = \widetilde{\boldsymbol{Q}}^\top\sqrt{\boldsymbol{M}^{-1}}\boldsymbol{M}\sqrt{\boldsymbol{M}^{-1}}\widetilde{\boldsymbol{Q}} = \widetilde{\boldsymbol{Q}}^\top\widetilde{\boldsymbol{Q}} = \boldsymbol{I}_m\,. \qquad\square$$

## B  The Case of the Softmax Transfer Function

Consider the case where the input examples are probability distributions $\{\boldsymbol{x}_i\in\Delta^{d-1}\}$ belonging to the $(d-1)$-simplex $\Delta^{d-1} = \{\boldsymbol{u}\in\mathbb{R}^d_+|\boldsymbol{u}^\top\boldsymbol{1}_d = 1\}$. The transfer function in this case corresponds to $f_{\mathrm{SM}} = $ softmax which induces the KL Bregman divergence $D_{F^*_{SM}}(\boldsymbol{u}, \boldsymbol{v}) = \sum_j(u_j\log\frac{u_j}{v_j}) - u_j + v_j$. Requiring $f_{\mathrm{SM}}$ to be inevitable imposes the constraint $\mathrm{dom}\,f_{\mathrm{SM}} = \mathbb{R}^d - \{\pm c\boldsymbol{1}_d, c\in\mathbb{R}_+\}$ as $f_{\mathrm{SM}}(\boldsymbol{u} + c\boldsymbol{1}_d) = f_{\mathrm{SM}}(\boldsymbol{u})$ for $c\in\mathbb{R}$. Thus, for the principal directions, we have $\boldsymbol{1}_d\notin\mathrm{CS}(\boldsymbol{V})$ where CS denotes column span. The above constraint can be easily incorporated into the QR decomposition on the Riemann-Stiefel manifold in Algorithm 1 when using the Householder method for the internal QR step.

The Householder method applies a series of *reflections* $\boldsymbol{P}_i = \boldsymbol{I}_m - 2\frac{\boldsymbol{b}_i\boldsymbol{b}_i^\top}{\boldsymbol{b}_i^\top\boldsymbol{b}_i}$ s.t. $\boldsymbol{b}_i\in\mathbb{R}^m$ and $i\in[n-1]$, on matrix $\boldsymbol{A}$ such that

$$\boldsymbol{R} = \boldsymbol{P}_{n-1}\boldsymbol{P}_{n-2}\ldots\boldsymbol{P}_2\boldsymbol{P}_1\boldsymbol{A}$$

is upper-triangular. The orthonormal matrix $\boldsymbol{Q}$ then can be written as

$$\boldsymbol{Q} = \boldsymbol{P}_1\boldsymbol{P}_2\ldots\boldsymbol{P}_{n-2}\boldsymbol{P}_{n-1}\,.$$

The following proposition shows that, in order to obtain $\boldsymbol{Q} \in \mathrm{St}_{d,k}^{(\boldsymbol{M})}$ from $\boldsymbol{A}$ s.t. $\boldsymbol{Q}^\top \boldsymbol{M} \boldsymbol{1} = 0$ using Householder reflections, it suffices to augment $\boldsymbol{A}$ from left by a column of all ones and apply Algorithm 1. The resulting matrix $\boldsymbol{Q}$ corresponds to the first factor when the first column is dropped.

**Proposition 2.** *Given $\boldsymbol{A} \in \mathbb{R}^{m \times n}$ s.t. $n < m$, let $\widetilde{\boldsymbol{Q}}, \widetilde{\boldsymbol{R}} = \texttt{RS-QR}([\boldsymbol{1}_m, \boldsymbol{A}], \boldsymbol{M})$ by applying Algorithm 1 using Householder reflections. The $\boldsymbol{Q}$ and $\boldsymbol{R}$ factors, $\boldsymbol{QR} = \boldsymbol{A}$ s.t. $\boldsymbol{Q} \in \mathrm{St}_{d,k}^{(\boldsymbol{M})}$ and $\boldsymbol{Q}^\top \boldsymbol{M} \boldsymbol{1}_m = 0$, can be obtained from $\widetilde{\boldsymbol{Q}}$ and $\widetilde{\boldsymbol{R}}$, respectively, by dropping the first column of $\widetilde{\boldsymbol{Q}}$ and the first row and column of $\widetilde{\boldsymbol{R}}$.*

*Proof.* The first reflection $\boldsymbol{P}_1$ of the Householder method transforms the first column of the input matrix into a unit vector $s_1 \boldsymbol{e}_1$ where $s_1 \in \mathbb{R}$. Thus, the first column of the resulting $\widetilde{\boldsymbol{Q}}$ term is just a rescaling of the first column of the input matrix. Thus, for the augmented matrix $[\boldsymbol{1}_m, \boldsymbol{A}]$, the first column of $\widetilde{\boldsymbol{Q}}$ corresponds to $r_1 \boldsymbol{1}_m$ with $r_1 \in \mathbb{R}$ s.t. $r_1^2 \boldsymbol{1}_m^\top \boldsymbol{M} \boldsymbol{1}_m = 1$. The remaining columns of $\widetilde{\boldsymbol{Q}}$, i.e., the matrix $\boldsymbol{Q}$, correspond to the column space of $\boldsymbol{A}$ minus the direction $\boldsymbol{1}_m$. Also, we have $\boldsymbol{Q}^\top \boldsymbol{M} \boldsymbol{Q} = \boldsymbol{I}_n$ and $\boldsymbol{Q}^\top \boldsymbol{M} \boldsymbol{1}_m = 0$. Note that the direction $\boldsymbol{1}_m$ is redundant for the softmax function, i.e., $\mathrm{softmax}(\boldsymbol{QR}) = \mathrm{softmax}(\boldsymbol{A})$ where the softmax function is applied to each column and $\boldsymbol{R}$ corresponds to the upper-triangular matrix by removing the first row and column of $\widetilde{\boldsymbol{R}}$. $\square$

## C  The Full Bregman PCA with Mean Algorithm

We provide the full procedure for the Bregman PCA algorithm with mean, including the case of the softmax function.

---

**Algorithm 3** Bregman PCA with Mean (Including Softmax)

---

**Input:** $\mathcal{X} = \{\boldsymbol{x}_i \in \mathrm{dom}\, F^* \subseteq \mathbb{R}^d\}$, number of components $k < d$
**Output:** dual mean $\boldsymbol{m} \in \mathrm{dom}\, F$, $\boldsymbol{V} \in \mathrm{St}_{d,k}^{(\boldsymbol{H}_F(\boldsymbol{m}))}$, $\{\boldsymbol{c}_i \in \mathbb{R}^k\}$
$\boldsymbol{m} \leftarrow f^*\!\left(\frac{1}{|\mathcal{X}|} \sum_i \boldsymbol{x}_i\right)$
initialize $\boldsymbol{V}$ and $\{\boldsymbol{c}_i\}$
**repeat**
    **for** $i \in [|\mathcal{X}|]$ **do**
        $\hat{\boldsymbol{x}}_i \leftarrow f(\boldsymbol{m} + \boldsymbol{V} \boldsymbol{c}_i)$
        $\boldsymbol{c}_i \leftarrow \boldsymbol{c}_i - \eta_a \boldsymbol{V}^\top (\hat{\boldsymbol{x}}_i - \boldsymbol{x}_i)$
    $\boldsymbol{V} \leftarrow \boldsymbol{V} - \eta_V \sum_i (\hat{\boldsymbol{x}}_i - \boldsymbol{x}_i) \boldsymbol{c}_i^\top$
**until** $\boldsymbol{V}, \{\boldsymbol{c}_i\}$ not converged
**if** $f$ is softmax **then**
    $\boldsymbol{V} \leftarrow [\boldsymbol{1}_d, \boldsymbol{V}]$                                       ▷ Augment a column of all ones.
$\boldsymbol{V}, \boldsymbol{T} \leftarrow \texttt{RS-QR}(\boldsymbol{V}, \boldsymbol{H}_F(\boldsymbol{m}))$
**if** $f$ is softmax **then**
    $\boldsymbol{V} \leftarrow \boldsymbol{V}_{:,1:}$                                    ▷ Drop the first column.
    $\boldsymbol{T} \leftarrow \boldsymbol{T}_{1:,1:}$                            ▷ Drop the first row and column.
**return** $\boldsymbol{m}, \boldsymbol{V}, \{\boldsymbol{T} \boldsymbol{c}_i\}$

---

## D   Matching Loss of a Transfer Function

Given an (element-wise) strictly-increasing (or cyclically monotone, in the more general case (Rockafellar, 1966)) transfer function $f : \mathbb{R}^d \to \mathbb{R}^d$, the *matching loss* between the (post-)target $\boldsymbol{y}$ and the prediction $\hat{\boldsymbol{y}} = f(\hat{\boldsymbol{a}})$ is defined as the line integral (Helmbold et al., 1999; Kivinen & Warmuth, 2001)

$$L(\boldsymbol{y}, f(\hat{\boldsymbol{a}})) := \int_{\boldsymbol{a}}^{\hat{\boldsymbol{a}}} (f(\boldsymbol{z}) - f(\boldsymbol{a})) \cdot \mathrm{d}\boldsymbol{z}, \tag{11}$$

where $\boldsymbol{a} = f^{-1}(\boldsymbol{y})$ is the pre-target. Note that the inverse function $f^{-1}$ always exists since $f$ is strictly increasing. By simplifying the integral, we can write the expression in Eq. (11) in terms of the convex integral function $F : \mathbb{R}^d \to \mathbb{R}$ s.t. $f = \nabla F$,

$$L(\boldsymbol{y}, f(\hat{\boldsymbol{a}})) = F(\hat{\boldsymbol{a}}) + F^*(\boldsymbol{y}) - \hat{\boldsymbol{a}} \cdot \boldsymbol{y} = D_{F^*}(\boldsymbol{y}, \hat{\boldsymbol{y}}) = D_F(\hat{\boldsymbol{a}}, f^{-1}(\boldsymbol{y})). \tag{12}$$

Eq. (12) shows the equivalence between the matching loss and the Bregman divergence induced by the convex integral function $F$ (or its Fenchel dual $F^*$). Also, the last expression indicates that $L(\cdot, f(\hat{\boldsymbol{a}}))$ is always convex with respect to $\hat{\boldsymbol{a}}$.

## E   Further Details on the Experiments

**CIFAR-10/100 Experiments:** For each experiment, we tune the learning rate and the momentum hyperparameters in the range $[10^{-5}, 0.2]$ and $[0.5, 0.99]$, respectively. The results are averaged over 5 independent trials for each experiment. The input images are normalized for each dataset by subtracting the mean and dividing by the standard deviation over the whole training set.

**Compute Resources:** For the CIFAR-10/100 experiments, we use V-100 GPUs. For the ImageNet-1k experiments, we use $8 \times 8$ TPU v3s.

## F   Experimental Results with Data Augmentation and Mixup

Data augmentation is a standard procedure for training deep neural networks. However, our main results in the paper focus on decoupling the information transfer from the teacher due to the knowledge distillation technique from the generalization improvement due to data augmentation. In this section, we showcase results using standard augmentation procedures (random horizontal flip and random crop) combined with Mixup (Zhang et al., 2018) with $\alpha = \beta = 1$ on the CIFAR-10/100 datasets. Table 2 presents the results of different methods with $k = 100$ components and 51 training epochs with a batch size of 128.

From the table, we see that the performance of all methods consistently improves with data augmentation. Additionally, among all methods, BRL performs the best. However, the difference between BRL and soft-label knowledge distillation is not as prominent as before (see Figure 4). Nonetheless, such behavior is not surprising as we expect soft-label knowledge distillation and our method to eventually reach the teacher's performance, given that a sufficient amount of data (i.e., training examples) for distillation is provided (Beyer et al., 2022). As we show in the main experiments, BRL delivers a clear advantage in a setup with a limited number of samples.

Table 2: Top-1 accuracy for different methods with number of components $k = 100$ and using data augmentation combined with Mixup (Zhang et al., 2018).

| Method | Dataset | |
|---|---|---|
| | CIFAR-10 | CIFAR-100 |
| Baseline | $83.65 \pm 0.41$ | $54.25 \pm 0.35$ |
| Baseline + Readout | $83.34 \pm 0.51$ | $55.11 \pm 0.34$ |
| Baseline Distillation | $86.00 \pm 0.22$ | $62.25 \pm 0.15$ |
| BRL | $\mathbf{87.03 \pm 0.12}$ | $\mathbf{62.72 \pm 0.28}$ |
| BRL + Readout | $86.24 \pm 0.29$ | $61.11 \pm 0.29$ |

