# OpenReview forum: "Layerwise Bregman Representation Learning of Neural Networks with Applications to Knowledge Distillation"
_TMLR — Accepted by TMLR_

### Review · Reviewer_CEhv · 2022-11-17

**Summary Of Contributions:**

This paper introduces a PCA algorithm with Bregman divergences under a generalized mean, which it utilizes to propose Bregman Representation Learning, a distillation method for approximating a large neural network with a smaller one. It demonstrates that this approach results in better predictive accuracies on image classification benchmarks (CIFAR, Imagenet) compared to some baselines, most notably knowledge distillation.

**Audience:**

Yes

**Claims And Evidence:**

No

**Requested Changes:**

Critical:
* Work out clear and explicit claims and discuss how they are supported by the theoretical and empirical results.
* Provide parameter counts for all methods as discussed in weaknesses.

Optional:
* I would suggest either editing this paper to be a more clearly focused neural distillation paper or adding discussion (ideally alongside empirical demonstrations) of other applications of the proposed method if the authors prefer to maintain their current general perspective.

**Strengths And Weaknesses:**

Strengths:
* Methodologically interesting, might be applicable more broadly (I would appreciate discussion on this though).
* Clear technical exposition and background discussion. I found that starting with an explicit and relatively verbose discussion of PCA effective in making the paper quite self-contained and easy to follow. I feel quite confident that implementing the method just based on the paper would be fairly straight-forward.
* The method appears to consistently perform better than the distillation baseline. There are large-scale results on ImageNet.

Weaknesses:
* No clearly stated claims. Essentially this paper is written like a conference submission ("we propose a new method and it works better than the baselines under certain metrics and datasets") rather than working out explicit claims and supporting them with evidence. E.g. "Our empirical findings indicate that our approach is substantially more effective for transferring information..." as in the abstract is extremely vague. I found this quite surprising as this is one of the two central acceptance criteria for this venue, although I will say that this is also my first time reviewing for TMLR, so I'm open to down-weighing this concern.
* Lack of high-level discussion. The introduction is essentially a technical background section, and provides no context on why this research is relevant/timely, what its overarching goal is or how the proposed approach is motivated.
* Related to the previous point, I find the positioning of the paper a bit odd. The title and abstract strongly emphasize the methodological contribution and argue that distillation is only an example application of the method, however there is no discussion of further applications (let alone demonstrations of these).
* The experiments do not discuss parameter counts at all, i.e. size of student network + $|m|, |V|$. My hunch is that for the proposed method they don't differ too much from classical knowledge distillation, however I haven't calculated this precisely -- these should really be provided in the paper. The storage complexity increases linearly in $k$, so I think these should be stated explicitly along all results (Figs 3 & 4, Tab 1). Given that (loosely speaking) the goal of distilling a larger network into a smaller one is to reduce storage/compute complexity under acceptable accuracy loss, I find it crucial to show some related metrics, parameter count would be the most obvious one.

Minor suggestions:
* In Fig 4, mark the distillation baseline with a horizontal line.
* I found the paragraph with the different variants at the bottom of page 10 a bit difficult to parse. Stating the hypothesis that you're testing with each approach/what ablation this corresponds to in (relatively) plain English would probably make it easier to read.
* I don't really follow the motivation for the cascade of student networks to distill each layer independently. Do you have reason to believe this would lead to an overall smaller approximation error while still being parameter efficient? I would like to see a little bit more detailed discussion on this and ideally a concrete experiment (can be exploratory, e.g. distilling individual layers vs entire ResNet blocks and comparing errors).
* Typo in Table 1, the first Accuracy column should say "Top-1 Accuracy" instead of "Top-5 Accuracy".

---

> ### Author Response · Authors · 2022-12-20
> **Response to Reviewer CEhv**
>
> Thank you for your comments. In the updated manuscript, we have applied the following changes.
>
> - We have improved the abstract and the presentation of the paper.
> - We moved the idea of a cascade of student networks to future directions for simplicity and provided clarification.
> We have provided parameter counts in Table 1.
> - "In Fig 4, mark the distillation baseline with a horizontal line": Please note that we use the same network (with an identical number of parameters) as the baseline. Thus, the baseline model becomes larger by increasing $k$ and, thus, cannot be represented as a horizontal line.
>
> The rest of the changes are discussed in the **General Comments**.

---

### Review · Reviewer_aEXP · 2022-11-20

**Summary Of Contributions:**

The authors propose an extension of Bregman PCA which estimates non-zero means and projection coefficients and can be extended to include constraints from non-linear activations. They then use Bregman PCA to compress and distill neural nets on the output layers. Their set of experiments demonstrate that the distillation performance is better than a soft-label baseline in terms of accuracy.

**Audience:**

Yes

**Broader Impact Concerns:**

Actually, some discussion of broader impacts, aka, “why do we care about this work?” would be helpful.

**Claims And Evidence:**

No

**Requested Changes:**

Major: the construction of QR in section 2.1 definitely should be renamed to something that is not “generalized QR”. “Generalized QR” is a separate technique for separately generating orthogonal matrices (see https://netlib.org/lapack/lug/node46.html for a quick introduction and [1] for more details). I’d suggest renaming this to something like “QR on the stiefel manifold”.

As a broader point, I believe that there should be some more discussion as to what exactly the transformed part of the QR decomposition is doing here. I think the transform $\sqrt{M} A$ is a retraction onto the manifold defined by $M$ but state this with low confidence. [2] may be helpful here.

Minor: spell out how you’d apply the constraints in either the more general case (rather than just softmax) and to a QR approach that isn’t simply householder (e.g. with Givens rotations, or realistically in the general case).

Minor: the paper would benefit from a definiation of the “convex integral function of the transfer function” – on pg 7.

Medium: while I quickly verified the algebra in Prop. 1, I’m a bit unclear about the intuition as to why the solution to the mean vector should just be the inverse map applied to the arithmetic mean? This also seems to assume function invertibility (which I see why there’s the restriction to monotonic transfer functions).

medium: Would we expect to see good results with transfer learning across datasets, or just good results from distillation?

Ideally, there’d be some discussion of “why” there’s better results on knowledge distillation? Is the optimization process made easier from this type of compression or is it mapping the data somehow? It seems like this intuition is a bit missing currently and should be explored a bit more, see [3,4] for some interesting metrics and experiments that can be studied.

Typos, etc:
-	Pg 6: change “decent” to descent
-	Pg 16: please just use two lines instead of the brackets in the proof of prop 1.
-	Pg 8: alg 1. “until … not converged”. I think you mean “while … not converged”
-	Pg 8: “we essentially minimize the matching loss of the transfer function”  unclear, are you doing this exactly or not?

References:
[1] Generalized QR Factorization and its Applications, Anderson, Bai, and Dongarra, Linear Algebra and Its Applications, 1992.

[2] Cholesky Based QR retraction on the generalized Stiefel manifold, Sato and Aihara, computational Optimization and its Applications, 2019.

[3] Does knowledge distillation really work?, Stanton et al, Neurips, 2021.

[4] Knowledge distillation: a good teacher is patient and consistent, Beyer et al, CVPR, 2022.



**Strengths And Weaknesses:**

Strengths:

I like the technical cleanness of the approach and of the modification to the QR algorithm.

The application of PCA to distillation is compelling.

The explanations of the technical parts are well explained overall.

Weaknesses:

The necessity of including the mean term in the PCA output is unclear and ill defined to me. In the other generalized PCA cases mentioned, it’s not especially limiting to assume a zero mean. In general, why can’t we just demean the features (or enforce that type of constraint during training)? There’s no space savings here because were storing a mean vector still?

The overall message of the paper is a bit confused – the experiments are written as if it was a distillation paper (but these experiments are limited), while the methodology is very heavy (as if it was a numerical analysis paper). What else can be done with this non-linear bregman PCA beyond just distillation of neural networks?

While not pitched as a “state of the art” paper, it seems very limiting to not compare to other reasonable compression / distillation approaches, for these experiments, e.g. adversarial ones [1] or other approaches [2,3]. I think the baselines of Coda PCA and simple soft label based distillation are a bit too weak for the lack of other experiments because it’s not clear if this is really a practical method for distillation / compression or just an interesting artifact of the math you did.

There’s no convergence analysis of the gradient descent algorithm for your approach – why should it converge to a stationary point? And, furthermore, the experiments mention an “online” version, why should this converge?

References:
[1] MEAL V2: Boosting vanilla Resnet-50 to 80%+ top-1 accuracy on ImageNet with-
out tricks., Shen and Savvides, 2020.
[2] Does knowledge distillation really work?, Stanton et al, Neurips, 2021.
[3] Knowledge distillation: a good teacher is patient and consistent, Beyer et al, CVPR, 2022.

---

> ### Author Response · Authors · 2022-12-20
> **Response to Reviewer aEXP**
>
> ### Including the mean term
> Similar to standard PCA, a zero mean assumption is invalid for non-centered data. In the Bregman generalization, we include a mean term to be consistent with the Euclidean counterpart. The memory overhead of including a mean term is negligible in practice. However, the mean term allows forming the Riemannian metric, thus the orthonormal $k$-frame, around the mean value. For neural networks, we agree that most activations are centered around zero because of the normalization at the input and the normalization layers such as BatchNorm. However, in the most general sense, including a mean vector significantly alters the geometry.
>
> ### Regarding the experiments
> Please note that the first part of the experiments is a PCA compression problem and thus, not necessarily limited to neural networks. In order to create a dataset with a non-flat geometry, we use the representation of a deep neural network. However, such datasets can be obtained by different means than neural networks.
>
>
> ### Comparison to other methods
> As pointed out by the reviewer, our goal is not to propose a SOTA distillation technique but rather to show the effectiveness of information transfer between models by using compressed representations. Our strategy can be trivially combined with several different distillation strategies, such as subclass distillation (Müller et al., 2020) and representation matching (Romero et al., 2014). The goal of the distillation experiment is to show that a significant amount of ''knowledge'' can be transferred by directly importing the learned representations.
>
> ### Convergence analysis
> We agree that convergence analysis is important for our method. However, such analysis is quite involved even for the Euclidean case and may fall outside of the scope of the current manuscript. However, note that the problem in Eq. (7) is convex with respect to $V$ and $c$ (not necessarily jointly convex), and thus an iterative block-coordinate descent approach with an appropriate step size is generally expected to converge to a stationary point.
>
> ### The inverse map for finding the mean
> The link function $f$ is the gradient of a strictly convex function $F$, and thus always invertible. (The more general case is known as cyclically monotone. See Appendix C.) For the softmax function, since the log-sum-exp function is not strictly convex along $\mathbf{1} \mathbb{R}$, we can consider the unique inverse function inv-softmax$(\mathbf{y}) = \log \mathbf{y} - \mathbf{1}\frac{1}{d} \sum_i \log y_i$.
>
> ### Transfer learning across datasets
> A common strategy for transfer learning is to pretrain a model on one dataset and fine-tune on another dataset. This strategy is shown to be effective, thus suggesting that neural networks learn "similar" representations of the datasets. By the same logic, we expect the overall Bregman representations to be transferable across datasets. However, we do not have empirical evidence at this point to support this claim.

---

### Review · Reviewer_YJiT · 2022-12-16

**Summary Of Contributions:**

This paper proposes using a variant of PCA—the so-called generalized Bregman PCA—as an approach for projecting neural network representations into lower dimensions. The generalized Bregman PCA method is the same as the original Bregman PCA, with the addition of a optimizable mean vector added to the reconstruction (resulting differences in the algorithm are listed in the introduction). The authors apply this method to distillation of neural networks, finding that their approach outperforms using vanilla PCA, and outperforms the common distillation method of training the student model to match the logits of the teacher model.


**Audience:**

Yes

**Claims And Evidence:**

Yes

**Requested Changes:**

- Generally speaking I would highly recommend revising Sec 2 and the latter part of Sec 1 to make the ideas and the new contributions (and their significance) clearer. For example, the following key points could be make much clearer: 1) what is the idea behind your algorithm for computing ‘generalized Bregman divergences’?, assume the reader doesn’t already know how to compute Bregman divergence PCA 2) why is this new generalized Bregman formulation a good idea for distilling neural network representations? What is it giving us above and beyond usual Bregman? Equivalently, why is it important to deal with non-centered data when we could just center data, compute Bregman PCA, then uncenter again? This could also be addressed with additional experiments comparing to this as a baseline.
- More specifically, remove the list on page 3, as it is not informative to the reader. Replace it instead with a schematic explanation of the new PCA method (projected gradient descent —> gradient computation easy and analytic, but how to project? Why does data lying on a different manifold (e.g., the simplex) require a change to the algorithm, and where is the change made?
- I hate to be that person asking for extra experiments. But it would be really nice if you could add runtime comparisons for your method versus e.g., usual PCA, and the Soft-label method in the case of distillation.

**Strengths And Weaknesses:**

First, thank you to the authors for pursuing an interesting direction, and bringing interesting mathematical ideas to the problem of model distillation. I see promise in this work and believe that it will of interest to the community. So I am recommending acceptance. However this verdict is very much giving the benefit of the doubt, and based on the TMLR philosophy that "only some readers need be interested" in order to justify acceptance. I have a number of concerns with the current manuscript that make it hard for me to determine the significance of the mathematical contribution, and hard to interpret the experimental results. I have attempted to give some critical, but constructive suggestions on how to set about framing this work more clearly.

---

> **Strengths**

- Introduction of a mathematically interesting variation on the Bregman divergence that is also allowed to minimize over a mean vector.
- The authors apparently closely follow the lines of prior Bregman divergence computation algorithms, making the necessary steps in order to re-derive an algorithm that also accounts for their desired mean vector.
- Method doesn’t appear to introduce additional hyperarameters above and beyond what were already needed for computing Bregman divergence.
- Method performs well against simple (yet sensible) baselines like PCA.

---

> **Weaknesses**

- The generalized Bregman PCA is introduced ‘to handle the case of non-centered data’. Clearly non-centered data is important, but what is wrong with just centering the data, computing the usual Bregman PCA, then uncentering again? Discussion of this point, or experiments showing it, would be very valuable to explaining the significance of this work.
- The contributions on page 3 are not made understandable. They are written as if the reader has already read the paper, as opposed to a the reader still trying to grasp the high-level contours. For instance in the first bullet point: why is it important that you added a generalized mean vector? Its it essential to have in order to formulate your distillation method, or just a nice-to-have? The second bullet: you “improve the orthonormality constraint… to conjugacy” - but what has this improved? Why is this better? And what is this used for? The third: what is QR decomposition and how is it related to the conjugacy constraint? A construction suggestion would be to include a high-level sketch of the algorithm structure (instead of only citing previous works Roy & Gordon and Chiquet).

---

> **General Assessment**

To conclude, this paper introduces a new Bregman PCA formulation and algorithm to compute it, which appears to be mathematically interesting. It appears to have some promising applications to distillation of neural networks - beating PCA alternatives, and the prototypical Soft-label Distillation method. Giving the benefit of the doubt I am therefore inclined to accept on the basis that this might be useful to some. However I would like to clearly register my main concern: the entire methods section is devoted to developing a ‘generalized’ Bregman PCA algorithm, following similar lines to the original. However nowhere in the manuscript (that I saw) was it explained or motivated why this generalized Bregman PCA was a better thing to be doing than usual Bregman PCA. And there are no experiments comparing generalized vs standard Bregman PCA. And I see no experiments comparing generalized vs standard Bregman. This leaves me quite uncertain as to what the value in the generalized method is, and whether perhaps existing Bregman divergence computation methods would work just as well. So I am quite skeptical, and if other reviewers have doubts about this paper then I would prefer to join a consensus to reject than to champion this papers acceptance.

---

### Author Response · Authors · 2022-12-20
**General Comments**

We thank the reviewers for their insightful comments and suggestions. It appears that most of the concerns and requested changes were editorial and involved the write-up, not the technical contributions of the paper. In particular, the reviewers find our approach "compelling" (Reviewer aEXP), "methodologically interesting, more broadly applicable" (Reviewer CEhv), "self-contained and easy to follow" (Reviewer CEhv), and declare that it brings "interesting mathematical ideas to the problem of model distillation", and "believe that it will of interest to the community" (Reviewer YJiT).

We have tried to address as many concerns as possible. The changes are marked in red in the updated draft. In particular, we address:
- **Exposition of the paper:** We outline our work as a layerwise representation learning method for deep neural networks with strictly monotonic transfer functions. Specifically, we emphasize the importance of representation learning and knowledge transfer in the abstract and the introduction. We then motivate our work as a way of exporting compressed representations from trained neural networks that has an application for knowledge distillation. We have rearranged the sections, moved some of the technical details to the appendix, and included additional references.
- **Importance of the mean vector for non-centered PCA:** In the standard PCA, including a mean vector is critical for handling non-centered data. The procedure amounts to subtracting the mean vector from the input and applying centered PCA. This procedure is identical to the case of generalized Bregman PCA: instead of subtracting the mean value from the input, the fixed mean vector enters the transfer function $f$ as a bias term. In the Euclidean case, this is equivalent to a subtraction as the function $f$ amounts to identity. Additionally, including a mean vector allows us to encapsulate our distillation method as adding a dense (or convolutional) layer to a student network's output with a fixed (trained) weight matrix $V$ and a bias vector $m$. We did not conduct additional experiments to analyze the effect of the mean due to the following reasons: 1) the effect of the mean vector is well-understood and studied extensively in the Euclidean case. The results apply to the Bregman case directly. 2) We use non-linear representations from trained neural networks for our experiments to replicate a real-world scenario. Neural networks, particularly ResNet models, produce almost centered activations due to the presence of BatchNormalization layers. Conducting additional experiments to evaluate the effect of mean involves synthetic data (as opposed to the previous case, with actual networks). Thus, such results are bland and do not add much to the paper's clarity.
- **Additional details about the experiments:** we have added the number of parameters and the runtime for our large-scale ImageNet-1k experiments.
- **Matching loss of a transfer function:** We have included a section in the appendix to describe the idea.


We hope our improved draft sufficiently addresses your concerns, and we look forward to hearing your feedback.

---

### Decision · Action_Editors · 2023-01-21

**Recommendation:** Accept as is

**Comment:**

The authors propose an extension of Bregman PCA, incorporating a mean vector and revising the normalization constraint on the principal directions. They then use Bregman PCA to compress and distill neural networks on the output layers.

The paper has a clear technical exposition and background discussion, as well as compelling experimental results on large-scale data.

Initial concerns by the reviewers about the clarity and scope of the work have been successfully addressed in the review period, which resulted in an extensive and constructive dialogue between the authors and reviewers. The revised paper now addresses technical details (e.g., highlighting the importance of information transfer between models as opposed to achieving SOTA distillation performance and other details, such as discussing the role of the mean vector for non-centered PCA).

Overall, this is a very interesting paper that will find its readership. I, therefore, recommend acceptance.

**Audience:**

Yes, the paper will be appealing to a decent fraction of the ML community.

**Claims And Evidence:**

All claims are sufficiently backed by evidence.

---

> ### Author Response · Authors · 2023-02-11
> **Camera ready version posted**
>
> We would like to thank the reviewers for their invaluable comments. We also thank the AE for the timely response and for coordinating the reviews and discussions. We have posted the camera ready version of the paper. Please let us know if there are any further comments or concerns.
>
> Regards,
> Authors